

# Uptake and toxicity of clothianidin to monarch butterflies from milkweed consumption

Timothy A. Bargar[1], Michelle L. Hladik[2] and Jaret C. Daniels[3,4]

[1] Wetland and Aquatic Research Center, U.S. Geological Survey, Gainesville, FL, USA
[2] California Water Science Center, U.S. Geological Survey, Sacramento, CA, USA
[3] Department of Entomology and Nematology, University of Florida, Gainesville, FL, USA
[4] Florida Museum of Natural History, Gainesville, FL, USA

## ABSTRACT

Recent concern for the adverse effects from neonicotinoid insecticides has centered on risk for insect pollinators in general and bees specifically. However, natural resource managers are also concerned about the risk of neonicotinoids to conservation efforts for the monarch butterfly (*Danaus plexippus*) and need additional data to help estimate risk for wild monarch butterflies exposed to those insecticides. In the present study, monarch butterfly larvae were exposed in the laboratory to clothianidin via contaminated milkweed plants from hatch until pupation, and the effects upon larval survival, larval growth, pupation success, and adult size were measured. Soils dosed with a granular insecticide product led to mean clothianidin concentrations of 10.8–2,193 ng/g in milkweed leaves and 5.8–58.0 ng/g in larvae. Treatment of soils also led to clothianidin concentrations of 2.6–5.1 ng/g in adult butterflies indicating potential for transfer of systemic insecticides from the soil through plants and larvae to adult butterflies. Estimated $LC_{50}$s for total mortality (combined mortality of larvae and pupae) and EC50 for larval growth were variable but higher than the majority of concentrations reported in the literature for clothianidin contamination of leaves.

## INTRODUCTION

Since their first development in the 1980s and initial registrations for use in agriculture in the 1990s, neonicotinoid insecticides have grown to comprise a major portion of the world's overall insecticide market with a wide variety of uses in crops, livestock, lawn and garden, and companion animals (*Jeschke et al., 2011*). They have replaced organophosphates and carbamates in agriculture because they are generally considered less toxic for nontarget organisms (*Uneme et al., 2006*; *Simon-Delso et al., 2015*). In the Midwestern United States, the neonicotinoid pesticides clothianidin and its precursor thiamethoxam are registered for use on corn and soybeans only as a seed coating. The high water solubility of these pesticides facilitates their absorption by the seedling upon germination and translocation throughout the plant where they are available for exposure to targeted herbivorous insect pests (*Simon-Delso et al., 2015*). In addition, the

Corresponding author
Timothy A. Bargar,
tbargar@usgs.gov

combination of high-water solubility, relatively long half-lives, and the fact that <20% of the neonicotinoids in the seed coating are taken up by target plants increases the likelihood of environmental contamination (*Hladik, Kolpin & Kuivila, 2014*; *Hladik, Main & Goulson, 2018*) as indicated by their presence in streams (*Hladik & Kolpin, 2016*), groundwater (*Bradford, Huseth & Groves, 2018*), and tap water (*Klarich et al., 2017*).

Recent concern for the adverse effects from neonicotinoid insecticides has centered on risk for insect pollinators in general and bees specifically (*Balfour et al., 2017*; *Woodcock et al., 2017*). Beekeepers began reporting in 2006 incidences of colony collapse disorder for honey bees. The collapses were occurring unexpectedly and have led to a variety of explanations including pesticides (*VanEngelsdorp et al., 2010*). While neonicotinoid insecticides are not the only pesticides to which the bees may be exposed, their role in the collapses has received considerable attention. The fact that bee colonies are used commercially to facilitate pollination of agricultural crops targeted by the neonicotinoid insecticides indicates an increased likelihood of exposure through pollen grains, which have been shown to contain the insecticides (*Rolke et al., 2016*; *Xu et al., 2016*). In addition, the *USEPA (2005)* noted a major risk concern for honey bees potentially exposed to clothianidin because of its toxicity to bees and recommended caution during applications. However, concern for the environmental impacts of neonicotinoids extend beyond the potential effects to bees (*Chevillot et al., 2017*; *Han, Tian & Shen, 2018*; *Hallmann et al., 2014*; *Morrissey et al., 2015*; *Zhang et al., 2018*).

Natural resource managers are concerned about the risk of neonicotinoids to conservation efforts for the monarch butterfly (*Danaus plexippus*). The long-term viability of North American populations is threatened (*Semmens et al., 2016*; *Schultz et al., 2017*; *Pelton et al., 2019*). The U.S. Fish and Wildlife Service was petitioned by The Center for Biological Diversity, The Center for Food Safety, and The Xerces Society in 2014 to list the monarch under the Endangered Species Act. Currently, the U.S. Fish and Wildlife Service is assessing the status of the population and the potential impact of multiple threats, including insecticides, to determine if listing is warranted. Simultaneously, monarch conservation practices and plans are being developed and enacted by States and multiple partners across the monarch's range to address threats and the immediate need to increase the population (*Commission for Environmental Cooperation (CEC), 2008*; *Caldwell, Preson & Cariveau, 2018*). Monarchs face a multitude of threats including loss of habitat quality throughout their breeding grounds due to insecticide use (*Stenoien et al., 2018*; *Crone et al., 2019*) including neonicotinoids (*Thogmartin et al., 2017*). The systemic nature of the neonicotinoid insecticides could lead to the contamination of milkweed plants and nectar resources. From this contamination, there can be dietary exposure potentially impacting monarch butterflies. The agricultural belt of the Midwestern United States is the breeding range for approximately 38% of the monarch butterflies that winter in Mexico (*Flockhart et al., 2017*) making the Midwestern United States a priority for monarch conservation efforts for the Fish and Wildlife Service. This region is also where neonicotinoid pesticide (thiamethoxam, imidacloprid, clothianidin) use is also high (*USGS, 2014*) increasing the exposure likelihood for monarch butterflies.

**Table 1 Dates and environmental conditions for Experiments 1, 2 and 3.**

| Experiment | Start date mm-dd-yyyy | End date mm-dd-yyyy | Temperature (Celsius) | | Relative humidity (%) | | Photoperiod (hh:mm) | |
|---|---|---|---|---|---|---|---|---|
| | | | Min | Max | Min | Max | Min | Max |
| 1 | 07-15-2018 | 08-20-2018 | 20.5 | 35.0 | 50 | 79 | 13:05 | 13:52 |
| 2 | 09-30-2018 | 10-30-2018 | 8.3 | 34.4 | 40 | 82 | 11:30 | 11:54 |
| 3 | 09-20-2019 | 10-11-2019 | 13.9 | 35.0 | 41 | 86 | 11:35 | 12:12 |

In the present study, we exposed monarch butterfly larvae to clothianidin, taken up by milkweed plants via application of a granular formulation to the soil, to evaluate its effect on monarch development. Monarch larvae were exposed from the time they hatched from eggs until pupation to the contaminated milkweed plants, and the effects of that exposure upon larval survival, larval growth, pupation success, and adult mass were measured.

## MATERIALS AND METHODS

Three separate experiments were conducted during this study. Experiment 1 evaluated butterfly response to dose levels based on label application rates for the pesticide product used in the experiments. Experiment 2 was conducted at higher dose levels due to low butterfly response in Experiment 1. Experiment 3 was conducted because aphids negatively affected the health of the control plants in Experiment 2, which likely influenced butterfly response in the controls. Information regarding the experiment start and end dates, temperature, photoperiod, and relative humidity is reported in Table 1. All experiments occurred at the laboratory facilities of the Wetland and Aquatic Research Center in Gainesville, Florida, USA (29.7261722° N, 82.4188444° W).

Swamp milkweed (*Asclepias incarnata*) was the host plant utilized in the experiments. The swamp milkweed is a local species and is grown pesticide-free by a local vendor. During Experiments 1 and 2, the plants were purchased from the local native plant vendor. The vendor is surrounded by fallow fields and pine plantations that are no nearer to the vendor than approximately 1 km. Due to heavy aphid infestation of those plants during the first two experiments, plants used in Experiment 3 were grown at the laboratory under controlled conditions. The laboratory-propagated plants were started from seeds (American Meadows, https://www.americanmeadows.com/) planted in organic seed-starter soil (Jiffy® Natural & Organic Seed Starting Mix). The planted seeds were placed under growth lights indoors. Two weeks after germination, the seedlings were transferred to larger pots and moved outside to be exposed to natural conditions (photoperiod, temperature, humidity) during the remainder of the growth period.

All plants were propagated in pots (one plant per pot) filled with organic potting soil (10% peat, 45% pine bark, 43% wood chips, 5% sand and 2% NutriHold™) instead of natural or artificial top soil to minimize the effect of top-soil compaction on plant health. A slow-release fertilizer was added to the potting soil before planting. All plants were watered every 2–3 days with enough water to moisten the soil, but typically not enough to result in water leaching from the pot. Weekly, the water was fortified with a liquid fertilizer

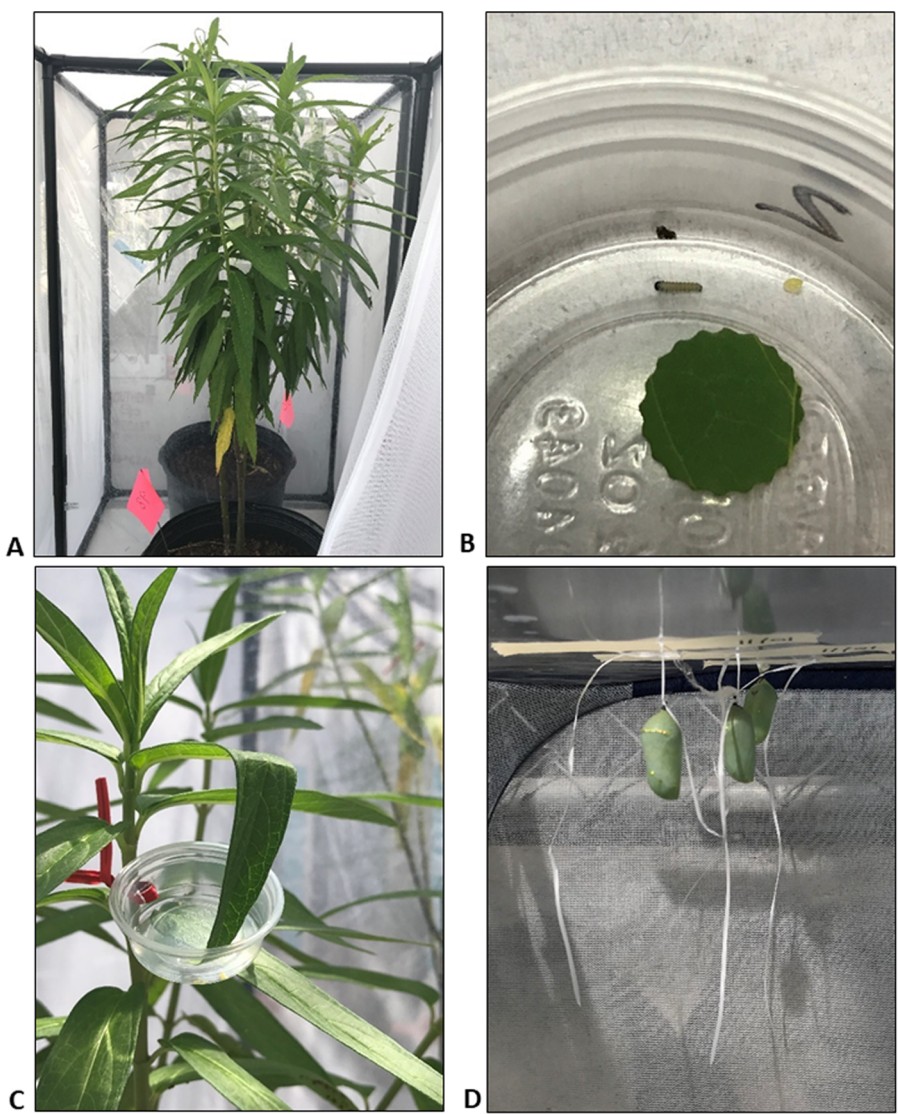

**Figure 1 Photographs of swamp milkweed (*Asclepias incarnata*) in screened terraria during Experiment 1 (A), newly hatched monarch larva in cup with 1 cm leaf disc from milkweed plant (B), cup attached to milkweed plant (C), and pupae harvested from one of the replicate terraria (D).**

(Miracle-Gro® Liqua Feed® 12:4:8 N–P–K ratio). All plants were maintained outside in screened terraria (42 cm deep × 76.2 cm wide × 121.9 cm tall; 1 plant per terrarium) before and during all experiments to prevent access of wild monarch butterflies and predators to the plants, and to retain larvae and adults during the experiments (Fig. 1A).

All terraria were maintained outside to ensure exposure to ambient conditions, but under a translucent roof to enable moisture control in the pots while ensuring a natural photoperiod and light intensity. After planting, each plant was assigned a unique number to be used in their random assignment (Microsoft Excel©) among dose levels for the experiments.

Monarch eggs were purchased from a local butterfly farm vendor (Shady Oak Butterfly Farm, https://shadyoakbutterflyfarm.com/). The colony is continuously maintained by the vendor for commercial purposes and is regularly supplemented by wild stock. The following procedures are taken by the vendor to ensure the colony is not infected by the *Ophryocystis elektroscirrha* parasite, which can adversely affect health and survival of larvae and pupae. All milkweed plants are raised indoors to eliminate deposition of *O. elektroscirrha* spores onto the colony plants by potentially infected wild adults; colony breeders are regularly checked for the presence of spores; all eggs are rinsed in a bleach solution to kill spores (*Altizer & Oberhauser, 1999*); and wild stock captured for introduction for infusion of additional genetic diversity into the colony are raised separately for several generations to monitor for eradicate *O. elektroscirrha* and disease.

The pesticide product used to dose the soils with clothianidin was Arena® 0.25G, which is a granular pesticide containing 25% clothianidin by weight. Arena® 0.25G is used in commercial and residential landscapes for control of a wide variety of herbivorous insect pests. Use of the granular product facilitated treatment of the soils with a known clothianidin mass without introduction of unwanted pesticides in the formulation that could be associated with the use of treated seed. The label rate of this product for flower beds (1.63 kg per 92.9 m$^2$, converted from the rate of 3.6 pounds per 1,000 square feet noted on the label). Since the soil surface area in each pot during Experiments 1 and 2 was 0.085 m$^2$, the label rate would be 1.5 g per pot for those experiments. Since smaller pots were used for Experiment 3 (soil surface area 0.0345 m$^2$), the label rate for those pots would have been 0.6 g per pot.

Experiment 1 consisted of 6 replicates in each of 5 dose levels plus a control. Three plants were randomly assigned to each replicate to ensure adequate host plant for the larvae (108 plants total evenly divided among 36 replicate sets). Plant height in Experiment 1 ranged from 0.6 to 1.2 m. The mean Arena 0.25G mass added to each pot was 0.08 (SD = 0.004), 0.21 (0.004), 0.64 (0.004), 1.92 (0.004) and 5.77 (0.006) g for dose levels 1, 2, 3, 4 and 5, respectively. No granular pesticide was added to the Control pots. All treated pots were dosed once on 15 June 2018 and then watered and fertilized as noted earlier. Monarch eggs were received on 18 July 2018 and divided among 36 pre-numbered plastic cups. The cups were randomly assigned to one of the three plants in each of the 36 replicates (one cup per replicate). When the eggs began to hatch, a single 1 cm diameter leaf disc from the associated plant was placed into the cup (Fig. 1B). A total of 24 h after the first egg within a cup hatched, the cup was attached to the associated plant to allow the larvae to crawl onto the plant (Fig. 1C). Larvae were transferred by hand from one plant to the next in a replicate when the source plant was nearly defoliated. No plants within a replicate were reused. The maximum number of larvae per replicate ranged from 1 to 8 depending on hatching success for the cup. The approximately 1 month delay between egg purchase and dosing ensured adequate time for clothianidin uptake by and distribution within the milkweed plants before introduction of the larvae to the plants. Each plant was inspected daily to monitor larval growth. The length of each larva found on a plant was measured and recorded daily (nearest 0.01 mm as determined by digital calipers). Mass was not utilized as the growth indicator to minimize handling stress

as a potential factor in larva growth. Dead larvae were noted and collected when found. Small larvae (<2 days of age) were most easily located by searching for forage areas on the leaves, but sometimes they were not found on plants with heavy foliage or with large numbers of aphids. Missing larvae (typically 1st instar) were not assumed dead because of the difficulty of finding small (1st and 2nd instar) larvae. The day post-hatch on which larvae began to pupate (prepupa) and on which pupae were first observed was recorded for each replicate. The pupae were monitored daily until eclosion (adult emergence from pupa), and the day post-pupation on which eclosion occurred was noted. The mass and forewing length (thorax to distal wing tip, to the nearest 0.01 mm) was measured for each adult butterfly. Prepupae were considered dead if they failed to completely form a pupa within 24 h, while pupae were considered dead if adult eclosion failed to occur.

Milkweed leaves, larvae, and adult butterflies were collected for clothianidin residue analysis. Composite leaf sampling was conducted to ensure that all milkweed plants that hosted larvae were sampled for analysis. Three composite leaf samples were collected from each dose level. The 6 replicates within each dose level (e.g., 1, 2, 3, 4, 5 and 6) were randomly divided into 3 pairs (e.g., 2 and 5, 1 and 6, 3 and 4) from which each composite sample was collected. The paired replicates were sampled twice (sampling A and B) during the experiment with the tissue from two samplings (A + B) comprising the composite sample. The first sampling (A) that occurred when larvae were <7 days old represented exposure of the early developmental stage. The A sample consisted of 3 × 1 cm leaf discs collected from each plant that had hosted larvae. The second sampling (B) that occurred when larvae were >7 days of age represented exposure of the later developmental stage. Because the larger larvae consume more leaf material and would be exposed to more pesticide, sampling B consisted of 6 × 1 cm leaf discs collected from each plant that hosted larvae. The locations on the plant from which leaf discs were collected were chosen arbitrarily, but when available, samples were collected from all portions (new growth to old growth) of the plant. All leaf discs from a replicate pair were composited within a Ziploc bag and placed into a freezer (−20 °C) until analysis. Compositing the first and second sampling resulted in 31 cm$^2$ of leaf sample with a mass that ranged from 0.173 to 0.306 g wet weight. All residue data for leaf tissues represent clothianidin concentrations among plants in the replicate. A single 5th instar larva (0.28–1.64 g per larva) and adult was collected for analysis from each of three randomly chosen replicates in each dose level and control. The larvae were placed into separate plastic culture tubes and placed into a freezer until analysis.

Based on the response observed during Experiment 1, a second experiment (Experiment 2) was conducted. Fewer plants were used in this experiment because the plants received from the vendor were in poor health due to heavy aphid infestation. The aphid populations were addressed before the experiment by both physical removal and soap water sprays, but the amount of time available to rehabilitate the plants was limited because of declining temperatures and photoperiod associated with the change in season (summer to fall). Both factors reduced plant growth and recovery meaning fewer plants suitable for the study were available for Experiment 2. As a result, the number of replicates was reduced to 3 per dose level and control. The dose rates in Experiment 2 were

higher relative to Experiment 1: 1.93 (SD = 0.01), 3.86 (0.004), 7.71 (0.006), 15.40 (0.004) and 30.08 (0.004) g of product per pot. Other differences relative to Experiment 1 include a shorter time period between pot dosing (4 September 2018) and monarch egg receipt (24 September 2018), and newly hatched larvae (2–7 per replicate) were placed onto random locations of the associated plant by using a camel-hair brush rather than by attaching the cup to the plant. Leaf sampling in Experiment 2 was like that for Experiment 1 with the exception that the samples were not composited among replicates. The composite leaf sample mass collected from each plant ranged from 0.266 to 0.363 g wet weight. All pupae were transferred from the terraria to a separate rearing cage to shelter them from the sun and heat. Pupae were moved by first tying dental floss to the cremaster, removing it from the attachment surface by gripping the silk pad with tweezers and pulling, and then taping the dental floss to the rearing cage interior ensuring the pupae hung naturally (Fig. 1D). In addition, not all larvae collected for analysis were in the 5th instar because larvae in some dose levels (4 and 5) died or went missing before attaining the 5th instar. As a result, some of the larvae collected for analysis were dead and much smaller (0.003 g) than those collected during Experiment 1. Also, no adult butterflies were available from Dose Levels 4 to 5, and only a single adult was available for Level 3. As a result, a single adult butterfly was chosen from each replicate that had adults during Experiment 2.

It should be noted that aphid populations on plants in the control group of Experiments 1 and 2 were very high most likely due to those plants being clothianidin-free. This impacted plant health, our ability to locate newly hatched larvae, and could affect larval behavior. The last noted impact could result in larvae having difficulty finding suitable foraging locations. While plants at the lower dose levels in Experiment 1 also had aphids, they were less abundant.

Due to concerns for the effect of aphids on results in the first two experiments, Experiment 3 was conducted to ensure the milkweed plants were not infested with aphids. Rather than purchase milkweed from the vendor, seeds were sown in a seed-starter soil (Miracle-Gro® seed starting potting mix) and placed under plant growth lights in the laboratory. Approximately 2–3 weeks after emergence, the seedlings were transplanted into larger pots containing a custom potting soil mix (described previously) and placed into terraria outside to expose the seedlings to natural environmental conditions (photoperiod, temperature, humidity) for continued growth. Watering and fertilization of the plants were conducted for these plants as described previously for Experiments 1 and 2. The plants were randomly assigned into the control group and 5 dose levels. The dose rate (g product per pot) was 0.78, 1.55, 3.11, 6.2 and 12.11 for Dose Levels 1–5, respectively. The rates for this experiment were lower relative to Experiment 2 due to smaller soil surface area for Experiment 3 (0.0345 m$^2$) compared to Experiment 2 (0.0856 m$^2$). Control plants were not treated with the granular pesticide product. Milkweed plant height and the number of leaf pairs per plant at the start of the experiment were 53 cm (SD = 5.2) and 33 pairs (SD = 5.2), respectively. As was done in Experiment 2, larvae were individually transferred to the plants at hatch. In contrast to the first two experiments, each replicate was comprised of one larva for each of two plants to ensure adequate availability

of food and reduce competition among larvae for available food. In addition, each larva was transferred to an apical leaf. Leaves were sampled at three time points during larval growth (first, middle and last instar) from all plants on which the larvae fed. Whole leaves were taken from the portion of the plant (top, middle, or lower third) on which the larva was feeding at the time and composited (3 leaves per composite) together for analysis. Most of the leaves were taken from the top third of the plants. Data for larval growth and pupation were collected as described previously in Experiments 1 and 2. No larvae or adults were collected for analysis due to the reduced number of larvae per replicate.

## Analytical procedure

All samples were weighed before analysis. Approximately 0.2–0.3 g of milkweed leaves or individual larvae and adults were dried/homogenized with sodium sulfate. Samples were spiked with a recovery surrogate (imidacloprid-d$^4$; Cambridge Isotope, Tewksbury, MA, USA) and extracted using an ASE$^®$ 200 (Dionex, Sunnyvale, CA, USA) using a 50:50 mixture of acetone:dichloromethane (1,500 psi; 100 °C). The extracts were solvent exchanged into acetonitrile and passed through solid-phase extraction cartridges containing 500 mg graphitized carbon (Restek, Bellefonte, PA, USA). The samples were then evaporated to 200 μL and spiked with an internal standard (clothianidin-d$_3$; Sigma–Aldrich, St. Louis, MO, USA). Extracts were analyzed on an Agilent 1260 bio-inert liquid chromatograph (LC) coupled to an Agilent 6430 tandem mass spectrometer (MS–MS). Instrument details are given in *Hladik & Calhoun (2012)*. The theoretical limit of detection (LOD) was 3 ng/g for the leaves (0.3 g sample) and 1 ng/g for the larvae and adults (1 g sample). All reported concentrations herein are for wet weight.

## Statistical analysis

Mortality and larval growth were the endpoints evaluated during this study. Length measurements for all larva in a replicate were averaged each day throughout larval development during Experiments 1 and 2. Larva length measurements for each replicate during Experiment 3 represent a single larva. A bivariate plot of larva age and length for each replicate in the dose levels and control yielded an independent indication of larval growth in response to dietary clothianidin exposure. Larval growth appeared to increase exponentially until larvae began to pupate at which time larvae in the replicate dropped out of the average size calculation. An exponential curve was fit (Microsoft Excel©) to growth data for larvae that had reached at least the 4th instar (13–25 mm length) to estimate the growth rate as described by the exponential growth equation $A_t = A_0 e^{kt}$ where $A_t$ is larva length at time $t$, $A_0$ is the intercept, $k$ is the growth rate, and $t$ is time in days. Since some larvae died during the early instars, the potential influence of clothianidin on larval growth was assessed by comparing growth rate for larger larvae (at least at the 4th instar) to clothianidin concentrations in leaves of plants consumed by the larger larvae. Growth data were initially compared by ANOVA followed by the post-hoc Holm-Sidak test ($\alpha = 0.05$).

Microsoft Excel© was used to estimate (by Probit Analysis, *Finney (1952)*) the LC$_{50}$ based on clothianidin concentrations in leaves. An LC$_{50}$ was estimated for larvae and for

total mortality (sum of mortality for larvae, prepupae and pupae) for Experiment 2. An LC$_{50}$ was estimated only for larval mortality in Experiment 3 since 100% mortality occurred in Dose Levels 2–5 not enabling estimation of an LC$_{50}$ for total mortality. No dose-response relation was evident for mortality or larval growth during Experiment 1. The numbers of larvae that were collected for residue analyses and that could not be found (i.e., missing larvae) were not included in the Probit analyses.

Residue levels in leaves and larvae, as well as adult mass and larval mortality, were compared among dose levels and control by ANOVA if the appropriate assumptions for the test were satisfied, or by the nonparametric Kruskal-Wallis test if the assumptions were violated. Where possible, the ratio of clothianidin concentrations in larvae and leaves (larva:leaf) was evaluated as a means to estimate clothianidin concentrations in larvae based on analysis of leaves. If the ratio is consistent regardless of exposure (i.e., concentrations in leaves), then the ratio may be a useful tool for estimation of larval exposure.

## RESULTS

### Experiment 1

The dose rates to soils in Experiment 1 did not result in measurable clothianidin concentrations in leaves in Dose Levels 1–3 (method detection limit or MDL = 4.3 ng/g), but did result in measurable concentrations in milkweed leaves in Dose Levels 4 (11 ng/g, SD = 3.6) and 5 (54 ng/g, SD = 27.2) (Table 2). Those concentrations did not differ significantly ($\chi^2 = 3.00$, $p = 0.083$, df = 1). Concentrations in larvae were also below detection (0.9 ng/g) in Dose Levels 1–3 and in adults were below detection (1.4 ng/g) in Dose Levels 1–4 (Table 2). The concentrations in larvae were marginally different between Dose Levels 4 and 5 ($\chi^2 = 3.857$, $p = 0.05$, df = 1). Two of the three adult butterflies from Level 5 analyzed for clothianidin had detectable concentrations (3.1 and 5.2 ng/g).

The relation between clothianidin concentrations in the leaves and effects in monarch butterflies was not clear in this experiment. Mortality of larvae and pupae was observed, but neither was proportional to clothianidin exposure (Fig. 2). Mean adult mass (F ratio = 0.59, df = 5, $p = 0.7$) and forewing length (F ratio 1.19, df = 5, $p = 0.35$) did not vary with clothianidin exposure and did not differ significantly among the dose levels and control (Table 3). The percentage of missing larvae varied among the dose levels and control but did not trend with clothianidin concentrations in leaves (Fig. 2). Only the percentage of larvae reaching the adult stage trended with clothianidin concentrations in leaves. However, the low percentage reaching the adult stage in the control may have resulted from the high percentage of missing larvae and dead pupae, both indicating stressful conditions for the larvae and pupae.

Larva growth for all dose levels and control increased from hatch until approximately 9 days post-hatch when larvae began to pupate (Fig. 3). The growth rates during that time (0.26–0.28, Table 4) did not differ significantly among the dose levels and control (F ratio = 0.7, df = 5, $p = 0.6$) and was not proportional to clothianidin concentrations in the leaves.

**Table 2 Clothianidin concentrations in leaves, larvae, and adults during the three experiments.**

| Dose level | Dose rate[1] (g/pot) | Clothianidin concentrations (ng/g) in matrices[2] (mean ± SD, $n$) | | |
|---|---|---|---|---|
| | | Leaves | Larvae | Adults |
| | Experiment 1 | | | |
| Control | 0 | BD | BD | BD |
| 1 | 0.08 | BD | BD | BD |
| 2 | 0.21 | BD | BD | BD |
| 3 | 0.64 | BD | BD | BD |
| 4 | 1.92 | 11 ± 3.6, 3 | 6 ± 3.3, 3 | BD |
| 5 | 5.77 | 54 ± 27.2, 2 | 13 ± 3.4, 3 | 3 ± 2.6, 3 |
| | Experiment 2 | | | |
| Control | 0 | BD | BD | BD |
| 1 | 1.93 | 20 ± 3.4, 3[a] | 5 ± 5.4, 3 | BD |
| 2 | 3.86 | 105 ± 80.4, 3[a,b] | 22 ± 16.3, 2 | BD |
| 3 | 7.71 | 107 ± 16.3, 3[a,b] | 58 ± 22.1, 2 | 5, 1 |
| 4 | 15.4 | 263 ± 152.5, 3[b,c] | 18 ± 13.5, 2 | NP |
| 5 | 30.08 | 843 ± 614.6, 3[c] | 20, 1 | NP |
| | Experiment 3 | | | |
| Control | 0 | BD | NL | NS |
| 1 | 0.78 | 54 ± 42.3, 6[a] | NL | NS |
| 2 | 1.55 | 91 ± 41.9, 4[a,b] | NL | NP |
| 3 | 3.11 | 232 ± 94.2, 5[a,b] | NL | NP |
| 4 | 6.20 | 1007 ± 461.3, 6[b,c] | NL | NP |
| 5 | 12.11 | 1545 ± 481.2, 4[c] | NL | NP |

Notes:
[1] Grams of product (clothianidin comprises 0.25% of product mass) added to the pot. Soil surface areas ($m^2$) in the pots was 0.0856 for Experiments 1 and 2, and was 0.0345 for Experiment 3.
[2] Significant differences between means were found for larvae in Experiment 1 ($\chi^2$ = 3.857, df = 1, $p$ = 0.05), and for leaves in Experiments 2 ($F$ ratio = 10.236, df = 4, $p$ = 0.001, post-hoc pair-wise comparison by Tukey's HSD) and 3 (Kruskal–Wallis test on the ranks $H$ = 20.593, df = 4, $p$ < 0.001, post-hoc pair-wise comparisons by Dunn's method with $\alpha$ = 0.05). Means with different superscript letters are significantly different. Significant differences were not found for the other matrices.
BD, below detection. Detection limit ~1 ng/g per gram of sample (leaf, larva and adult mass (g) ranged from 0.173 to 0.781, 0.003 to 1.644 and 0.426 to 0.977, respectively); NP, no pupae in dose level; NL, no larvae available for sampling; NS, adults not sampled.

## Experiment 2

The greater clothianidin dose to soils during this experiment resulted in detectable concentrations in leaves and larvae from all dose levels (Table 2). Average clothianidin concentrations in leaves were 20 (SD = 3.4), 105 (80.4), 107 (16.3), 263 (152.5) and 843 ng/g (614.6) for Dose Levels 1, 2, 3, 4 and 5, respectively (Table 2). Clothianidin concentrations in larvae did not differ significantly among the dose levels (Kruskal–Wallis statistic 6.255, df = 4, $p$ = 0.18), but they did peak at 58 ng/g when concentrations in leaves averaged 107 ng/g (Dose Level 3) and decline at the higher dose levels despite higher concentrations in leaves (Table 2). Detectable clothianidin concentrations in adults were

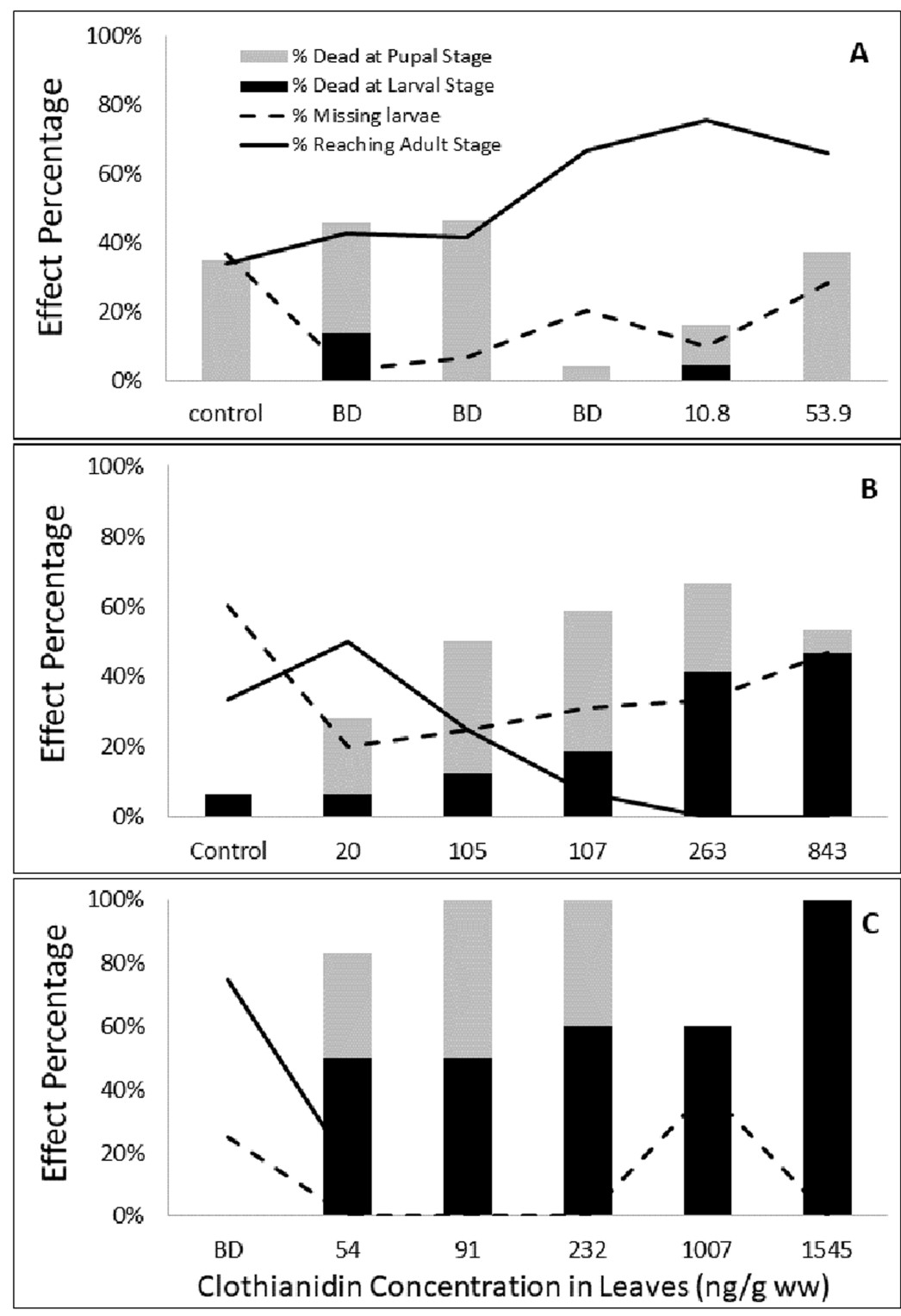

**Figure 2 Effect of clothianidin concentrations in leaves on larval and pupal survival during Experiments 1 (A), 2 (B) and 3 (C).** The sum of % dead (larvae and pupae), % missing and % reaching adult stage is 100%.

**Table 3 Larval growth rates and adult size in relation to clothianidin concentrations in leaves.**

| Dose level | Clothianidin in leaves (ng/g)[1] (mean ± SD) | Adult mass (g)[2] (mean ± SD, $n$) | Forewing length (mm)[3] (mean ± SD, $n$) |
|---|---|---|---|
| | Experiment 1 | | |
| Control | BD | 0.65 ± 0.197, 4 | 47.4 ± 0.37, 4 |
| 1 | BD | 0.75 ± 0.168, 5 | 46.7 ± 5.33, 5 |
| 2 | BD | 0.73 ± 0.118, 5 | 50.3 ± 3.48, 5 |
| 3 | BD | 0.71 ± 0.186, 6 | 49.4 ± 3.02, 6 |
| 4 | 10.8 ± 3.2 | 0.80 ± 0.111, 5 | 50.8 ± 0.73, 5 |
| 5 | 60.3 ± 22.2 | 0.69 ± 0.035, 4 | 49.3 ± 0.64, 4 |
| | Experiment 2 | | |
| Control | BD | 0.51 ± 0.068, 3 | 49.4 ± 1.70, 3 |
| 1 | 20.2 ± 3.4 | 0.67 ± 0.065, 3 | 49.9 ± 1.18, 3 |
| 2 | 105.4 ± 80.4 | 0.57 ± 0.012, 3 | 46.2 ± 6.14, 3 |
| 3 | 107.2 ± 16.3 | 0.52, 1 | 47.0, 1 |
| 4 | 177 ± 52.8 | NP | NP |
| 5 | 155 | NP | NP |

**Notes:**
[1] Mean clothianidin concentration in leaves of plants consumed by larvae that had grown to at least the 4th instar (≤15 mm in length).
[2] Adult mass did not differ significantly among the dose levels and control for Experiments 1 ($F$ ratio = 0.59, df = 5, $p$ = 0.7) and 2 (Kruskal–Wallis statistic = 5.6, df = 2, $p$ = 0.061).
[3] Forewing length did not differ significantly among the dose levels and control for Experiments 1 ($F$ ratio 1.19, df = 5, $p$ = 0.35) and 2 (Kruskal–Wallis statistic = 1.156, df = 2, $p$ = 0.561).
NP, no pupae in dose level.

found for only Level 3 (107 ng clothianidin/g leaf)—no adults eclosed in Level 4 (263 ng/g leaf) and no pupae were formed in Level 5 (843 ng/g leaf).

Larval and total mortality (larval plus pupal mortality) were inversely proportional to clothianidin concentrations in leaves (Fig. 2). The $LC_{50}$ for total mortality (47 ng/g, 95% CI [29.3–75.8]) was lower than the $LC_{50}$ for larval mortality (205 ng/g, 95% CI [117.4–357.0]). The lower total mortality $LC_{50}$ (mortality at the larval and pupal stages) indicates that pupal stage may be more sensitive to clothianidin exposure than larval stage.

A relation was also evident between clothianidin concentrations in leaves and both the percentage of missing larvae and the percentage of larvae reaching the adult stage (Fig. 2). The percentage of missing larvae increased from 20% to 47% in Dose Levels 1 (20 ng/g leaf) and 5 (843 ng/g leaf), respectively. However, the high percentage of missing larvae in the control (60%) indicates a factor other than clothianidin may have contributed to larval disappearance. The percentage of larvae reaching the adult stage declined from a high of 50% in Dose Level 1 to 0% in Dose Levels 4 and 5 (263–843 ng/g leaf, respectively). The lower percentage in the control (33%) is likely a reflection of the high percentage of missing larvae in the control.

Neither adult mass (Kruskal–Wallis statistic = 5.6, df = 2, $p$ = 0.061) nor forewing length (Kruskal–Wallis statistic 1.156, df = 2, $p$ = 0.561) differed significantly among dose levels and control (Table 3). Relative to Experiment 1, larvae growth was not as uniform among the dose levels (Fig. 3). Larvae in the Control and in Dose Levels 1–3

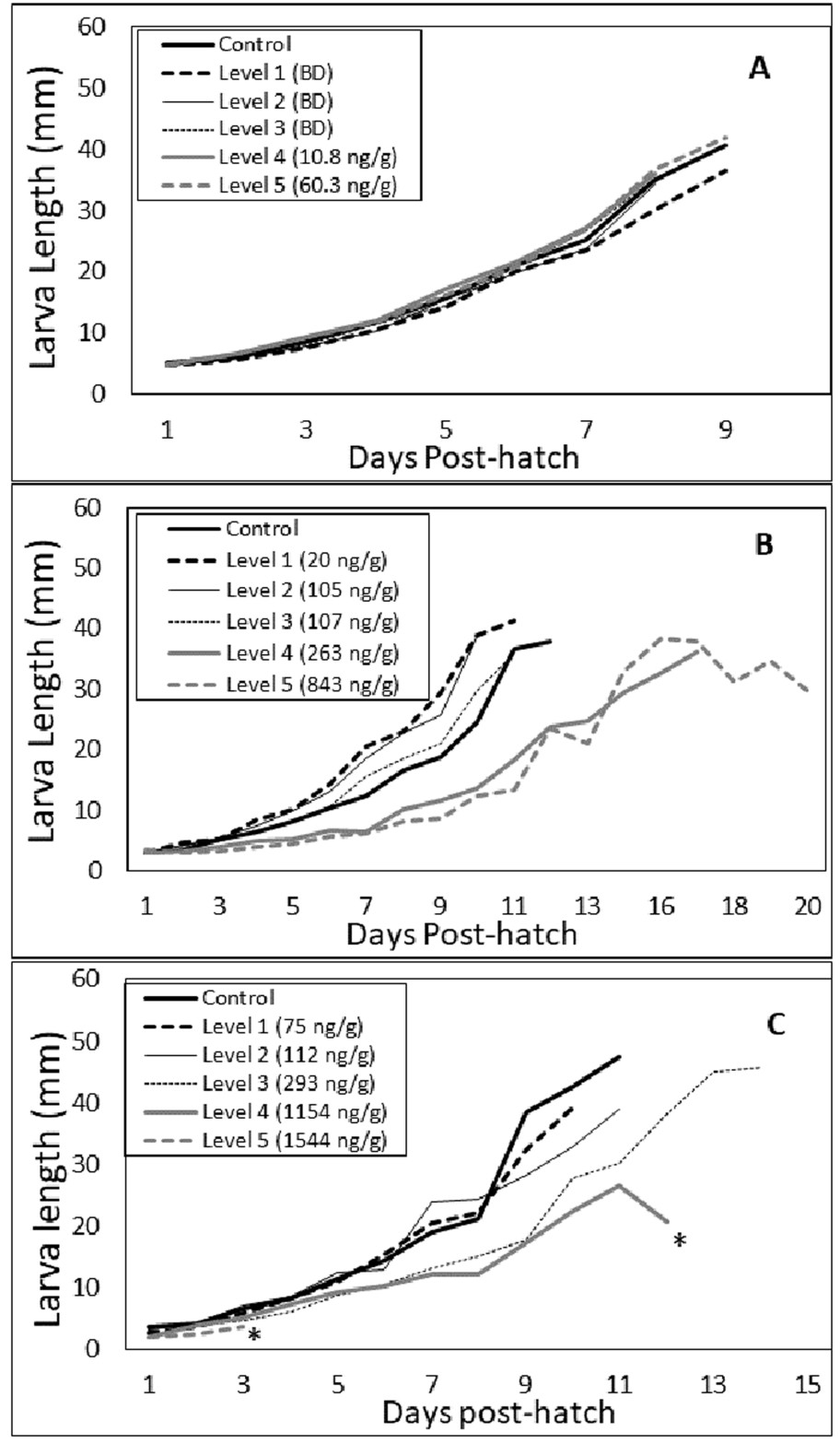

**Figure 3 Larvae growth during Experiments 1 (A), 2 (B) and 3 (C).** The shown lines are based on the mean larval length for replicates within the respective dose level or control and terminate on the last day prior to formation of pupae in the dose level or control. The concentrations shown in the legends indicate the mean for clothianidin in the leaves from all plants on which larvae had fed. Growth rates for larvae in

**Figure 3** (continued)
Experiment 1 did not differ significantly among dose levels ($F$ ratio = 0.492, df = 5, $p$ = 0.8), but they did differ significantly in Experiments 2 ($F$ ratio = 5.7, df = 4, $p$ = 0.014) and 3 ($F$ ratio = 4.6, df = 3, $p$ = 0.037). Asterisks shown in Experiment 3 denote dose levels (4 and 5) in which no larvae survived to the pupal stage.

(20–107 ng/g leaf) grew faster than larvae in Dose Levels 4 and 5. In fact, the growth rates declined significantly ($F$ ratio = 5.7, df = 4, $p$ = 0.014) from Dose Levels 1–5 (Table 4). Relative to larvae in the control group, growth of larvae consuming leaves with an average clothianidin concentration up to 155 ng/g (Dose Level 5) was not significantly reduced (Table 4). Clothianidin concentrations in leaves of 177 ng/g (Dose Level 4) did significantly reduce larval growth ($p$ = 0.034, Holm-Sidak post-hoc comparison) (Table 4).

Clothianidin concentrations in the different matrices collected from a dose level consistently declined from leaves to larvae to adults (Table 2). This consistency indicates estimation of larval body burdens based on concentrations in leaves may be possible based on a ratio of concentrations between larvae and leaves (larva:leaf). The ratio ranged from 0.14 to 0.74 in Experiment 1, averaging 0.42 (SD = 0.230), and from 0.04 to 0.8 in Experiment 2, averaging 0.30 (SD = 0.233). However, a bivariate plot of the ratio relative clothianidin concentrations in leaves revealed that the ratio was inversely proportional to clothianidin concentrations in the leaves (Fig. 4). The inverse relationship indicates that the larva:leaf ratio is not independent of clothianidin concentrations in the leaves, and that a single ratio would not be sufficient to estimate larval contamination. The decline of clothianidin in larvae despite increasing dietary contamination may also mean larvae are consuming less food as clothianidin contamination increases, which was indicated by the lower growth rates (Table 4; Fig. 3).

## Experiment 3

Average clothianidin concentrations in leaves during Experiment 3 were 54 (SD = 42.3), 91 (41.9), 232 (94.2), 1,007 (461.3) and 1,545 ng/g (481.2) for Dose Levels 1, 2, 3, 4 and 5, respectively (Table 2). Those concentrations increased with dose and differed significantly ($\chi^2$ = 20.593, df = 4, $p$ < 0.001) (Table 2). Clothianidin concentrations in leaves in Dose Levels 4 (1,007 ng/g) and 5 (1,545 ng/g) were higher than in leaves from Dose Level 5 of Experiment 2 (843 ng/g). No adults were analyzed since 3 of the 4 were from the Control group while the 4th was from Dose Level 1 in which clothianidin residues, based on results from Experiments 1 to 2, would likely have been undetectable.

Larval consumption of the contaminated leaves negatively affected monarch butterflies. Larval mortality ranged from 50% in Dose Levels 1 (54 ng/g leaf) and 2 (91 ng/g leaf), to 60% in Dose Levels 3 (232 ng/g leaf) and 4 (1,007 ng/g leaf), to 100% in Dose Level 5 (1,545 ng/g leaf) (Fig. 2). Between 33 (Level 1) and 50% (Level 2) of the monarch butterflies died at the pupal stage. The $LC_{50}$ for larval mortality (66 ng/g, 95% CI [9.7–451.4]) was lower than the $LC_{50}$ for larvae during Experiment 2 (205 ng/g) but approximated the $LC_{50}$ for total mortality in Experiment 2 (47 ng/g). An $LC_{50}$ for total mortality was not estimated for Experiment 3 since partial mortality was observed only at Dose Level 1

**Table 4 Monarch larvae growth in relation to clothianidin concentrations in leaves.**

| Dose level | Growth rate ($k$)[1,2] (mean ± SD, $n$) | Intercept ($A_0$) (mean ± SD) | Clothianidin in leaves[3] (ng/g ± SD) |
|---|---|---|---|
| | Experiment 1 | | |
| Control | 0.26 ± 0.040, 5 | 4.08 ± 0.746 | BD |
| 1 | 0.26 ± 0.026, 6 | 3.67 ± 0.632 | BD |
| 2 | 0.26 ± 0.021, 6 | 3.65 ± 0.893 | BD |
| 3 | 0.28 ± 0.022, 6 | 3.57 ± 0.668 | BD |
| 4 | 0.27 ± 0.032, 6 | 3.98 ± 0.707 | 10.8 ± 3.2 |
| 5 | 0.27 ± 0.025, 5 | 3.87 ± 0.872 | 60.3 ± 22.2 |
| | Experiment 2 | | |
| Control | 0.24 ± 0.023, 3[a,b] | 2.35 ± 0.155 | BD |
| 1 | 0.27 ± 0.007, 3[a] | 2.57 ± 0.1258 | 20.2 ± 3.4 |
| 2 | 0.26 ± 0.029, 3[a] | 2.53 ± 0.1903 | 105.4 ± 80.4 |
| 3 | 0.23 ± 0.042, 3[a,b] | 2.60 ± 0.1046 | 107.2 ± 16.3 |
| 4 | 0.17 ± 0.035, 2[c] | 2.71 ± 0.663 | 177 ± 52.8 |
| 5 | 0.16, 1[b,c] | 2.63 | 155 |
| | Experiment 3 | | |
| Control | 0.27 ± 0.027, 3[a] | 2.86 ± 0.170 | BD |
| 1 | 0.27 ± 0.035, 3[a] | 2.54 ± 0.199 | 74.7 ± 53.8 |
| 2 | 0.27 ± 0.035, 3[a] | 2.58 ± 0.375 | 111.8 ± 11.6 |
| 3 | 0.22 ± 0.018, 4[a,b] | 2.54 ± 0.244 | 355.6 ± 152.8 |
| 4 | 0.17, 1[b] | 3.19 | 1,153.9 |
| 5 | – | | |

**Notes:**
[1] Larval growth rate modeled by the exponential growth equation $A = A_0 e^{kt}$ where $A$ is the estimated larva length at time $t$, $A_0$ is the intercept, $k$ is the growth rate, and $t$ is the day post-hatch. Growth rate estimated only for larvae that had grown to at least the 4th instar (≤15 mm long).

[2] Mean larval growth rate was significantly different among dose levels and control in Experiments 2 ($F$ ratio = 5.7, df = 4, $p$ = 0.014) and 3 ($F$ ratio = 4.6, df = 3, $p$ = 0.037), but not in Experiment 1 ($F$ ratio = 0.7, df = 5, $p$ = 0.6). Means within Experiments 2 and 3 with different superscript letters are significantly different based on post-hoc comparisons (Holm-Sidak method). No larvae in Dose Level 5 of Experiment 3 survived to the 4th instar, meaning no growth rate was determined for that dose level.

[3] Mean clothianidin concentration for leaves of plants consumed by larvae that had survived to at least the 4th instar. The means do not include contamination in plants consumed by larvae that died in the earlier instars (Instars 1–3). BD, below detection. Detection limit ~1 ng/g per gram of leaf sample (leaf sample mass ranged from 0.173 to 0.781 g).

(Fig. 2). Larval mortality reduced the number of larvae that reached the adult stage from a high in the Control group (75%) to 17% in Dose Level 1 and 0% in the higher dose levels.

Larval growth (Fig. 3; Table 4) differed significantly among dose levels ($F$ ratio = 4.6, df = 3, $p$ = 0.037). Growth was unaffected by clothianidin up to a concentration of 355.6 ng/g in leaves but was affected at 1,154 ng/g (Dose Level 4). No larvae in Dose Level 5 reach the pupal stage. Only four butterflies successfully eclosed in this experiment, three in the control and one in Dose Level 1 (54 ng/g leaf).

No data were collected for the adult butterflies since there was only a single adult from the treated replicates.

All data generated from the present study and used in the preparation of this paper can be accessed at https://doi.org/10.5066/P9QX4OJW.

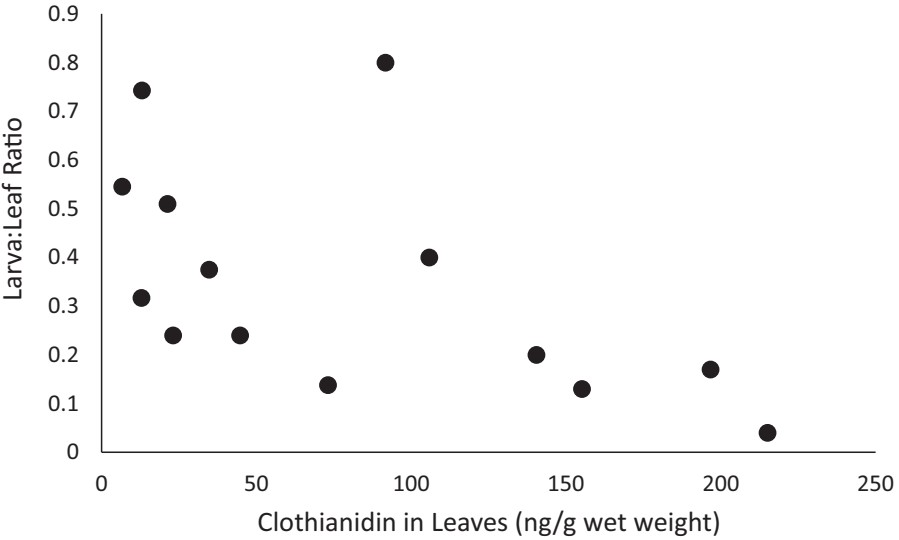

**Figure 4 Relation between clothianidin concentrations in leaves and the ratio of clothianidin concentrations in larvae and the leaves (Larva:Leaf ratio) during Experiments 1 and 2.**

## DISCUSSION

The goal for this study was to estimate effect levels relating clothianidin residues in milkweed leaves to adverse effects in monarch butterflies. Such data would assist resource managers focused on the risk of neonicotinoid contamination in general and clothianidin specifically to monarch butterflies. Some investigations have reported on the correlations between increased neonicotinoid use and butterfly population trends (*Gilburn et al., 2015*; *Forister et al., 2016*), but they are limited in their utility for resource managers concerned with monarch butterfly conservation because they did not directly evaluate neonicotinoid toxicity. Several studies have directly investigated neonicotinoid toxicity to lepidopterans and reported adverse effects in nontarget (*Krischik et al., 2015*; *Pecenka & Lundgren, 2015*; *Basley & Goulson, 2018*; *Whitehorn et al., 2018*) and target (*Ding et al., 2018*) lepidopterans. Three of those studies report dietary exposure levels that have utility for risk estimation (*Basley & Goulson, 2018*; *Ding et al., 2018*, *Krischik et al., 2015*). One evaluated clothianidin toxicity for monarch butterflies (*Pecenka & Lundgren, 2015*), but it did not report dietary exposure levels. The goal for the present study was estimation of the dietary exposure level to clothianidin associated with adverse effects in monarch butterflies.

Three separate experiments were conducted to attain this goal. The first experiment found no significant effect (larval growth and survival, adult mass, pupal survival) at mean clothianidin concentrations in leaves up to 59 ng/g wet weight. This concentration in leaves is generally greater than what has been reported in the literature. *Bredeson & Lundgren (2019)* reported concentrations of 0.6–9.7 ng/g in leaves of cover crops interseeded with thiamethoxam-treated corn seed crops; *Main et al. (2017)* reported clothianidin concentrations from below detection (0.002–0.66 ng/g)—2.01 ng/g in plants

from wetlands bordering treated canola crops; *Botías et al. (2016)* reported concentrations ranging from below detection (detection limit = 0.12 ng/g) to 11.45 ng/g in foliage of plants bordering treated oilseed rape fields; *Basley & Goulson (2018)* reported clothianidin concentrations ranging from below detection (0.2 ng/g)—48 ng/g in plants from the margins of clothianidin treated wheat fields; and *Olaya-Arenas & Kaplan (2019)* reported concentrations in milkweed plants (median less than the detection limit, 1.06 ng/g) ranging from below detection—56.5 ng/g. The results from these and Experiment 1 of the present study indicate that clothianidin concentrations in wild milkweed plants are generally not high enough to adversely affect monarch butterflies. However, the amount of data on clothianidin concentrations in nontarget plant foliage is limited.

The second and third experiments were conducted at a higher dose level to increase the likelihood of finding an exposure level associated with observable effects; Experiment 3 was conducted to eliminate the possible effect of aphids during Experiment 2. The elevated exposure in both experiments led to adverse effects on survival and growth. Lethal effect levels ($LC_{50}$ for clothianidin in leaves) ranged from 47 to 205 ng/g, while larval growth was reduced at 177 ng/g during Experiment 2 and 1,153.9 ng/g during Experiment 3. These results are in line with those reported by others. *Basley & Goulson (2018)* reported that larva size for the common blue butterfly (*Polyommatus icarus*) was affected at a relatively low clothianidin concentration in leaves (14.5 ng/g). Results from the present study indicate no effect on monarch larval growth at that exposure level. However, the same study reported that 439.1 ng/g in leaves significantly decreased larval survival, an exposure level higher than the $LC_{50}$ values estimated in the present study. *Ding et al. (2018)* estimated a dietary $LC_{50}$ of 27.77 μg/g (27,700 ng/g) for larvae of a pest moth species (black cutworm, *Agrotis ipsilon*). That value was much greater than the dietary $LC_{50}$ values estimated in the present study and by *Basley & Goulson (2018)*. *Pecenka & Lundgren (2015)* reported an $LC_{50}$ of 15.6 ng/g for monarch larvae, but it was unclear if that value represents concentration in leaves or in the solution used to dose those leaves.

While effect levels were estimated for monarch butterflies based on data from the present study, the application rates in the Experiments 2 and 3 exceeded those recommended on clothianidin product use labels. Those application rates resulted in residue levels in the leaves largely higher than what has been reported in the literature, but that were useful in generating exposure levels to achieve effects on the measured endpoints and estimate effect levels.

During the present study, the application rates were 9.4–681.7 g a.i./acre (grams of active ingredient (clothianidin) per acre) for Experiment 1 and 228.0–3,553.8 g a.i./acre for both Experiments 2 and 3. No effect was observed at rates less than the label rate, during Experiment 1, neither were they evident at higher rates. The second and third experiments were conducted at greater application rates to determine the residue level in the plants that leads to an adverse effect. The rates in the Experiment 2 and 3 were much greater than those noted on the labels for several pesticide products (Arena® 0.25G, Poncho®, Belay®, Sepresto 75 WS, Arena® 50 WG, Clutch™ 50 WDG) with clothianidin (4.5–180.7 g a.i./acre). The results indicate monarch butterflies may be relatively insensitive to clothianidin at label application rates, clothianidin availability from potting

soil may be low relative to that typical for top soil, or a combination of the two were needed to elicit effects.

High aphid density on some milkweed species can induce production of the toxin cardenolide by the plant (reviewed in *Agrawal et al. (2012)*), and *Zalucki, Brower & Malcolm (1990)* reported early instar monarch larvae are negatively affected by high cardenolide concentrations. Since the milkweed plants in the control and low dose groups of Experiment 1 and in the control group of Experiment 2 suffered from high aphid densities, the aphids may have indirectly affected the monarch larvae. Indeed, larval growth rate in the control group of Experiment 2 was low relative to the controls in the other two experiments, but it was not significantly lower. On the other hand, the percentage of missing larvae in the controls for Experiments 1 and 2 (40–60%)—was an issue. However, toxin production by swamp milkweed, the species used in this study, is not induced by high aphid densities (*Martel & Malcolm, 2004*; *Zhender & Hunter, 2007*). Therefore, the effect of aphids on the results of this study may be behavioral resulting in the larvae dropping from the plant leading the high percentage missing in the control plants.

To our knowledge, this is the first study to present data demonstrating that soil clothianidin residues can result in clothianidin exposure for adult butterflies through larval consumption of contaminated leaves. *Krischik et al. (2015)* conducted a similar study in which they tested the possibility that adult butterflies and lady beetles may be affected as a result of feeding from flowers contaminated by imidacloprid translocated from soil. While lady beetle survival was affected, they reported no significant mortality for adult monarch or painted lady (*Vanessa cardui*) butterflies exposed to flowering milkweed plants (*Asclepias curassavica*). While residue levels in butterflies were not measured in that study, the lack of effect in the butterflies should not be assumed to indicate no exposure since exposure was clearly adequate to affect lady beetles. However, exposure for the lady beetles and butterflies was presumed in that study rather than confirmed through analyses. Exposure of adult butterflies in the present study as a result of larval consumption of leaves was assessed by analyses of larvae and adults. There is no indication of bioaccumulation of clothianidin, but clothianidin is not completely metabolized and eliminated by larvae allowing its transfer to adults. Also, while adult butterflies can be captured and analyzed for clothianidin, doing so may not be the best method in field studies to determine butterfly exposure to systemic insecticides. That is because in the present study, either adverse effects occurred at earlier life stages when residues were not detectable in adults, or adverse effects at earlier stages prevented complete development to the adult stage. The concentrations detected in the adult butterflies would represent a sublethal exposure, the toxicity of which is unknown. To our knowledge, the sublethal toxicity of clothianidin for adult butterflies has not been reported. *Whitehorn et al. (2018)* reported sublethal toxicity (reduced size) of the neonicotinoid insecticide imidacloprid for adult farmland butterflies (*Pieris brassicae*) exposed during the larval stage. The present study did not find adult mass or forewing length were significantly affected by larval consumption of clothianidin contaminated leaves.

Neonicotinoid insecticide mobility in soils is inversely related to soil organic carbon (*Zhang et al., 2018*; *Singh et al., 2018*). *Zhang et al. (2018)* reported that sorption affinity of clothianidin for four different soils declined with organic carbon content, while *Singh et al. (2018)* reduced clothianidin concentrations in the leachate of soils by amending them with farm yard manure. The organic carbon content for the potting soil used in the present study was measured to be approximately 80%. The soil organic matter content in natural top soils of the Western (~1%, *Derner, Augustine & Frank, 2019*) and Northern Great Plains (~3%, *Frank et al., 1995*) are much lower, and likely represent the expected organic content of soils on which the application rates for clothianidin are based. Therefore, availability of clothianidin from the potting soil was likely significantly reduced in the present experiment compared to what would be expected for natural top soils where this pesticide is typically applied. As a result, the effects observed during the present study are better associated with clothianidin concentrations in the leaves as opposed to the soil application rates.

## CONCLUSIONS

The present study was conducted to determine the effect of dietary clothianidin exposure upon monarch butterfly development. This study is the first to report the transfer of a clothianidin from soil to adult monarch butterflies only from larvae exposed through consumption of contaminated leaves. Monarch butterfly survival and larval growth were adversely affected by increased clothianidin concentrations in leaves. But the clothianidin concentration leading to those effects was variable with $LC_{50}$s ranging from 47 to 205 ng/g and effects upon growth at 177 and 1,154 ng/g. The measured effects levels were greater than most of the data for clothianidin concentrations in leaves reported to date. When combined with data from field monitoring of clothianidin concentrations in milkweed leaves, those effect levels will aid in risk estimation for monarch butterflies exposed to that neonicotinoid in the field. Further research is needed to determine if toxicologically relevant doses can be transferred from larvae to adult butterflies.

## ACKNOWLEDGEMENTS

We would like to express our gratitude to Samm Epstein who assisted with most of the growth measurements, Marc Godts of Green Isle Gardens who supplied the milkweed plants and provided advice on care for the plants, and Shady Oak Butterfly Farm who provided the monarch eggs. Any use of trade, product, or firm names is for descriptive purposes only and does not imply endorsement by the U.S. Government.

### Funding

Funding for this research was provided by U.S. Fish and Wildlife Service and the U.S. Geological Survey. The funders had no role in study design, data collection and analysis, decision to publish, or preparation of the manuscript.

## Grant Disclosures

The following grant information was disclosed by the authors:
U.S. Fish and Wildlife Service.
U.S. Geological Survey.

## Competing Interests

The authors declare that they have no competing interests.

## Author Contributions

- Timothy A. Bargar conceived and designed the experiments, performed the experiments, analyzed the data, prepared figures and/or tables, authored or reviewed drafts of the paper, and approved the final draft.
- Michelle L. Hladik conceived and designed the experiments, performed the experiments, analyzed the data, prepared figures and/or tables, authored or reviewed drafts of the paper, and approved the final draft.
- Jaret C. Daniels conceived and designed the experiments, performed the experiments, analyzed the data, prepared figures and/or tables, authored or reviewed drafts of the paper, and approved the final draft.

## Data Availability

All data generated are available at ScienceBase: Bargar, T.A., 2019, Uptake and toxicity of clothianidin to monarch butterflies from milkweed consumption (ver. 2.0, January 2020): U.S. Geological Survey data release, DOI 10.5066/P9QX4OJW.

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
