# Peer review of "Uptake and toxicity of clothianidin to monarch butterflies from milkweed consumption"

_PeerJ, doi:10.7717/peerj.8669_

## Round 0.1 · original submission · Major Revisions

Please clearly address the suggestions made by the reviewers. I look forward to receiving your revised manuscript.

Reviewer 1 ·

Basic reporting

Professional language is used throughout the manuscript. The introduction is reasonable, but some areas of clarification are needed.

Line 48 and other locations: Clarify that clothianidin is not registered for foliar applications on corn or soybean, which are the use sites most relevant for assessing monarch risks, at least in the Midwestern summer breeding range. Given the nature of the study, the results are primarily relevant to soil drench, soil-applied/granular or seed-treatment formulations/applications. The study design is marginally relevant for foliar applications. If the authors wish to ‘extrapolate’ their findings to foliar applications, it may be appropriate in the Discussion section.

Line 53: Suggested edit: “Also because of its high-water solubility, there is the potential for clothianidin to move downslope from a crop field with shallow ground and surface water runoff, which makes it available for uptake by non-target plants and potential exposure to animals that utilize those plants (Hladik, Main & Goulson, 2018).”

Line 80: Insecticides may degrade habitat in agricultural settings … While the Oberhauser et al 2006 paper is relevant for mosquito adulticide applications, it is not a relevant citation for the application and use site scenario in the current study. In addition, Anderson et al. and Dively et al address Bt toxin exposure, which is a significantly different story than soil applied/seed treatment formulation of a chemical insecticide. However, without citing these papers, the case can easily be made that testing the hypothesis that soil applied/seed treatment uses of clothianidin could cause adverse effects is reasonable given the existing, published studies documenting detection of clothianidin and other neonics in milkweed growing in and around corn/soybean fields. Also, here, or lines 96 – 98, or in the discussion also reference the study by Krischik et al 2015 (PLoS ONE 10(3): e0119133. doi:10.1371/journal.pone.0119133), who assessed effects of imidacloprid on monarch larvae (and adults) in a manner similar to this study.

Line 88: Provide a citation supporting near 100% use (it’s correct – but suggest citing Tooker and Douglas)

Line 89: what does ‘high’ mean? More accurate to say: ‘exposure cannot be precluded’

Line 102: soil application – be specific; i.e.., via granular soil application.

Experimental design

Line 109: Confirm distance between nursery and ag fields that could be using foliar or soil applied insecticides in the area.

Line 114-115: Provide here or later in the methods section, the average height of the plants, and number of leaves per plant at the start of each experiment. This information facilitates interpretation of the design and study results (i.e., the extent of potential bias in treatments – not withstanding random assignment of plants to treatments).

Line 132 – 133: Confusing sentence. Do the authors mean they didn’t want to plant clothianidin treated corn or soybean seeds in the pots since the resultant variability in soil concentrations would likely be extreme given the corn/soybean plants would also be absorbing the active ingredient etc etc? Totally reasonable approach for this study, but the sentence isn’t clear.

Lines 147 – 152: The aphid infestation creates a confounding factor(s). Even with treatment prior to initiation of Experiment 2 the plants were likely stressed and their nutritional status compromised. During the study the control larvae were likely experiencing greater adverse effects due to the aphids than the larvae on the treated plants. While this confounding issue does not preclude publication of the study, it does need to be addressed in the discussion section with regard to the uncertainty in the results and the generalizability of the findings.

Lines 162 – 163: For Experiments 1 and 2 provide additional detail as to where the larvae were placed on the plants. Were locations selected randomly, or were locations evenly dispersed on the bottom 1/3, middle 1/3, top 1/3 and new vegetation when 4 or more larvae on a plant? If less than 4 larvae how were they placed? This information probably relevant in terms of interpretation of the residue data – residue concentrations may be different based on the age of the leaf. Also females tend to lay eggs on newer vegetation, presumably due to higher nutritional content. But if multiple larvae (e.g., 4 to 8) are placed in close proximity on younger vegetation, there may not be sufficient biomass per larvae (as compared to a plant that was infested with 2 or 3 larvae), which could elicit intraspecific competition and or increased on or off plant movement of larvae. If some larvae were placed on older leaves, and less nutritional value, they may have a higher likelihood of mortality independent of clothianidin exposure.

Line 175: Was the condition of the plants also recorded daily and/or at the point larvae on a plant were all dead or missing or when they reached the pupal stage; i.e., number leaves with feeding, number of total leaves without feeding)? Was there evidence of feeding on plant leaves even when larvae went missing? This information could help assess the extent to which higher larval infestation rates did or did not create a potential bias due to limited biomass – especially with the 4th and 5th instars.

Line 179 – Missing larvae. Missing control larvae are likely dead assuming they didn’t escape the cages and a 4th or 5th instar or pupa didn’t ‘suddenly appear’ near the end of the study. Also double-check the raw data. For example, it seems in Experiment Two, 7, 2, and 6 larvae were placed on the three Treatment 4 plants; however, the number missing were 0, 4, and 3, respectively (i.e., more larvae were missing than originally assigned to the second plant).

Line 203: The leaf sampling isn’t clear. For Experiment 2, was a composite leaf sample collected for each of the 18 plants? Are residue results plant-specific?

Line 218: Is this method the same or similar to previous methods use by Hladik et al.? If so provide a reference or two.

Lines 249, 286, 295: Provide additional details as to the methods used to calculate dose-response relationships (i.e., what software package and code: R, drc?). Was Abbott’s formula used to correct for control mortality? Presumably, mortality rates per plant were used as inputs to the dose-response analyses. Confirm mortality events were based on observation of dead larvae. In addition to calculating a LC50 based on the sum of mortality across larvae, prepupae and pupa, also calculate the cumulative larval LC50 to enhance the means to discern differences in life stage susceptibility. As noted above, it would also be appropriate to calculate the dose-response relationships assuming missing larvae are dead (for control and treated plants) to assess to what extent that assumption influences the results.

As noted by the SD values in the text and data in the supplemental files, the variability of leaf concentrations is very high for levels 2 (196.6, 44.7, 75 ng/g), 4 (433.8, 140.4, 215.1 ng/g) and 5 (155.1, 1036.8, 1337.6 ng/g). In addition, the mean values for levels 2 and 3 are essentially the same. Is there any (qualitative) relationship between the location of sampled leaves (e.g., bottom 1/3, middle 1/3, top 1/3, new vegetation) and residue concentration? Using the mean leaf concentrations in the probit analysis masks the variability in the data. With Level 4 plants mortality (with missing larvae ‘dropped’), rates of 100%, 50%, and 50% is associated with 155.1, 1036,8, and 1337.6 ng/g, respectively. Assuming the residue data for Experiment 2 in the supplemental file are associated with specific plants (e.g., the three concentrations are associated with 3 specific plants) it would be best to consider each of the treated plants as an independent unit. The LC50 analysis would then use each plant’s specific ng/g leaf concentration and morality rate (i.e., 18 points total – 3 control points and 15 points for the treated plants).

Validity of the findings

Line 269: Claiming no control mortality is tenuous. As noted above, missing larvae in the control groups is likely mortality under the assumption a missing larvae did not re-appear in later instars/pupal stage. Perhaps there were some escapes from the cages, but it is unlikely all the missing larvae were escapes. Based on the data in the spread sheet for Experiment One controls: 3/5, 0/2, 2/2, 3/5, and 0/3 went missing. Suggest the dose-response analysis be re-done assuming missing = dead to see how much dropping the missing vs. assuming missing = dead influences the findings. Same for Experiment 2.

Line 307: The trends with the clothianidin treatments is noteworthy. However, it is difficult to interpret in light of the control data despite the author’s reasonable interpretation that the condition of the control and low dose plants was significantly compromised by the aphids. It is difficult to untangle the aphid X treatment interaction and should be clearly stated in the discussion.

Paragraph starting on line 324: Given the low number of samples and the variability of the plant residue levels care must be taken to not over interpret the significance of these data. It is reasonable to use the data to indicate there was a complete exposure pathway in the toxicity studies and that the relationship between leaf and larval residues is not inconsistent with the existing body of literature that indicates neonics do not bioaccumulate.

Paragraph starting on line 340: Include a summary of https://www.frontiersin.org/articles/10.3389/fevo.2019.00223/full
that was published after the current manuscript was submitted. In the paragraph do not use ‘up to’ concentrations. Only providing upper limit concentrations is a misleading summary of the published data given the skewed distributions of residues in most papers. Instead provide the % of milkweed samples with detects, the median concentrations. and the range for plants with detections. A summary along these lines provides a more complete and transparent sense of the residue data and how that data relates to the results of this study.

Paragraph starting on Line 357: Include a summary/comparison to results reported by Krischik et al 2015 (PLoS ONE 10(3): e0119133. doi:10.1371/journal.pone.0119133) for imadicloprid.

Line 376: Concur this is a reasonable summary of the Pecenka and Lundgren (2015) paper.

Line 380: While these may be the first data showing trophic transfer is it all that surprising since this compound is designed to kill piercing/sucking and chewing insect pests? The data does provide confirmation that there was clothianidin exposure to the larvae in the toxicity studies. The fact that the adults have a clothianidin load is not all that surprising, unless the authors assumed the pupae are highly efficient metabolizing clothianidin and would degrade all the residues accumulated through the 5th instar. I am not aware of any published data that would support such a hypothesis.

Lines 389, 392: If the authors wish to raise this issue, even though it is outside the domain of the paper (nectar concentrations were not measured in this study and other than successful eclosion no assessment of adult exposure was undertaken), they should cite and summarize the findings in Krischik et al 2015 (PLoS ONE 10(3): e0119133. doi:10.1371/journal.pone.0119133) for imadicloprid.

Paragraph starting on Line 397: This paragraph is stretching beyond what the data in this paper and the literature supports. What are the median residue levels in nectar? And what data exists to suggest an adult concentration of 5 ng/g is of toxicological concern? It would be reasonable to conclude that while this study captures most of the monarch life cycle but the potential effects of neonic exposure on adult survival, behavior, and reproductive performance have yet to be studied (with a note that some work done by Krischik et al.).

Paragraph starting on Line 426 can be deleted. It doesn’t add significant information/interpretation of this study’s results.

Paragraph starting on line 435 should be placed after the paragraph ending on line 424.

Additional comments

This study was challenging to undertake and the authors are congratulated for their efforts. Undertaking an ecotoxicity study that attempts to mimic an environmental setting, but maintain some degree of control over variables, is very difficult. This study will contribute to the evolving understanding of the potential risks of neonic exposure to monarchs; however, the paper requires significant revision before it can published. While it may not be possible to fully address the confounding variables and experimental design constraints, the revised paper needs to provide a more complete and transparent representation of the uncertainty in the results and their limitation for use in assessing risk.

·

Basic reporting

References:
The authors missed citing information. They should review the following studies:
- Jesse and Obrycki 2003. Occurrence of Danaus plexippus L. (Lepidoptera: Danaidae)
on milkweeds (Asclepias syriaca) in transgenic Bt corn agroecosystems
- Jesse and Obrycki 2003. Field deposition of Bt transgenic corn pollen: lethal effects on the monarch butterfly
- Hellmich et al. 2001. Monarch larvae sensitivity to Bacillus thuringiensis- purified proteins and pollen
- Hellmich et al. 2001. Impact of Bt corn pollen on monarch butterfly populations: A risk assessment.
- Wolt et al. 2003. A Screening Level Approach for Nontarget Insect Risk Assessment: Transgenic Bt Corn Pollen and the Monarch Butterfly (Lepidoptera: Danaidae)
- Krischik et al. 2015. Soil-Applied Imidacloprid Translocates to Ornamental Flowers and Reduces Survival of Adult Coleomegilla maculata, Harmonia axyridis, and Hippodamia convergens Lady Beetles, and Larval Danaus plexippus and Vanessa cardui Butterflies
- Mattila et al. 2005. Response of Danaus plexippus to pollen of two new Bt corn events via laboratory bioassay
- Oberhauser and Rivers. 2003. Response of Danaus plexippus to pollen of two new Bt corn events via laboratory bioassay.
- Obrycki et al. 2001. Transgenic Insecticidal Corn: Beyond Insecticidal Toxicity to Ecological Complexity

Line 165: provide a reference to justify waiting for 1-month between plant dosing and introduction of larvae.

Experimental design

Methods: Additional details and clarifications should be provided in the methods. Specifically:
1. Information about the location where the experiments where performed, the average temperature, humidity, photoperiod, and time of the year when the study was performed. This information should be provided for each experiment. Insecticides can degrade depending on environmental conditions.
2. How did the researcher ensure that the plants were not exposed to other pesticides due to drift? The authors should provide clarification on the methods.
3. Provide more information about the colony used for the experiments (i.e., the origin of the colony, the number of generations in the lab, is the colony supplemented with field insects?).
4. Where the adults checked for OE? If not, please provide a reference or clarification that rinsing eggs in bleach is enough to eliminate OE.
5. Line 130: indicate the commercial use of Arena 0.25G.
6. To increase the ability of other researchers to replicate the study, it will be useful to include a picture or diagram of the terrarium set up used.
7. Line 144-146: indicate in the rationale for the second experiment (e.g., .estimate LC50).
8. Specify what was used as the control.
9. Line: 71: Where the larvae transferred at the same time? Please clarify.
10. Lines 167-172: explain why fewer eggs were used in the second experiment.
11. Sample collection: indicate the conditions at which the samples were collected and stored. Is not clear what the author meant by randomly paired.
12. Line 208-209: indicate the tango of the total number of larvae evaluated per treatment per experiment.
13. Data analysis: explain why the difference of analysis was performed between experiment 1 and 2. Indicate the software(s) used for the analyses. Why not perform pairwise comparisons? More details need to be provided for the probit analysis, where the 95% confidence intervals calculated, slope and chi-square estimated? Was mortality corrected to control mortality? An EC50 could have been estimated with the growth rate.

Validity of the findings

Conclusions: the authors missed to discuss and speculated in without indicating that this was speculation and not a direct finding of the study. Specifically:
1. Plants with heavy aphid infestation can increase cardenolide levels in the plants. The authors should discuss if this could have affected the results of their experiments.
2. Line 343-344: Here is important to indicate the concentration found in the study was potentially higher due to the mode of application granular vs seed treatment or foliar, and formulation evaluated (i.e., different adjuvants will affect the effect of the product). The authors should expand on the discussion on the potential difference in concentration depending on the type of application and formulation.
3. Line 359-363: Authors should indicate that the results obtained in this study should be taken with caution given the limited sample size.
4. Lines 393-394: indicate the type of behavioral effects observed in bees and what this would imply for monarchs.
5. Lines 397-407: Authors should specify and discuss that the effects observed in this study were at concentrations higher than what is expected in the field. Also, they should discuss how the concentrations obtained in this study relate to spray applications since they infer that it will be a risk during flowering. Different types of applications/formulations will affect exposure in the field. Furthermore, in what type of landscape the product tested is expected? the effect of a product used in an orchard should not be assumed to have the same effect than in a cornfield for example. The authors should be careful with their conclusions given the limited sample size and the use of a laboratory colony and indicate the limitations of the study.

Figures: provide sample size in the figure legends. Figure 2 and 6 are hard to interpret change the line types or colors to increase legibility.

Additional comments

This paper reports the effects of milkweeds treated with granular clothianidin on monarch butterfly larvae from hatch to pupation, and the effects upon larval survival, larval growth, pupation success, and adult mass. The manuscript is well written, and the research is timely, relevant, and fills a critical knowledge gap. However, before it can be published the manuscript needs additional details and clarification in the methods, further discussion, and interpretation based on the specific use of the product tested, and revise some of the statements provided since they are highly speculative given the sample size evaluated and can lead to misinterpretation. Review the attached document for additional minor suggestions.

·

Basic reporting

There are a couple of locations where references should be added as indicated in the comments to authors. Other than that, the basic reporting in the article is fine.

Experimental design

A couple of locations where the Methods should be clarified regarding the data collection and analysis as indicated in the comments to authors.

Validity of the findings

A couple of locations where the authors might consider an alternative to data analysis and presentation of results in the Results section to improve the manuscript.

Additional comments

L 24 – please indicate here that this was a laboratory or controlled study and not a field study. Perhaps “…monarch butterfly larvae were exposed in the laboratory to contaminated…” would be suitable.
L 30 – if possible, please indicate some statistic or metric to your last sentence to provide evidence regarding reduced growth rates.
L 55 – change “utilize” to “use”
L 59-69 – might consider removing most of the second paragraph regarding bees and merging the topic sentence (L 58-59) with the next paragraph on L 71.
L 120-123 – Perhaps re-write this sentence to ensure it is clear as: “All eggs collected by the vendor from their breeding stock were rinsed in a bleach solution to ensure no infection by the OE (Ophryocyctis elektroscirrha) parasite from wild monarchs butterflies, which can result in mortality of larvae and pupae”. Should also cite an appropriate paper to support this such as: Altizer and Oberhauser 1999 Effects of the protozoan parasite Ophryocystis elektroscirrha on the fitness of monarch butterflies (Danaus plexippus). Journal of Invertebrate Pathology 74:76-88.
L 189 – males generally have larger wings than females so if you did not record sex at eclosion it would be good to note this here.
L 194-216 – thank you for writing this section so clearly. Very easy to understand and I appreciate that.
L 241 – would the value for A0 be zero or is A0 the length the day after hatching? If the later, then please state this.
L 243-246 – Can you indicate why you used different statistical tests for Experiment 1 and Experiment 2?
L 248-252 – Does the probit analysis provide the estimate of the LC50? If so, could you explain that further here as it wasn’t evident to me. If not, the LC50 analysis needs to be included in the Methods section.
L251-252 – Perhaps state this sentence first before explaining the analysis for data from Experiment 2.
L 254-256 – I think this is the information I was seeking in point above L 243-246. Can you somehow move the text in L 254-256 up to address that point?
L 261-264 – The text is not clear here. Perhaps better to have 1 or 2 sentence to present the results from leaves. Then 1 or 2 sentences to present the results from larvae. The last two sentences from adults is clear.
L 273 – Please change “Average” to “Mean”. Also remove the words “like mortality” in this sentence.
L 277-281 – I think this paragraph should be revised in a couple of ways. First, you didn’t test if the growth was exponential, you assumed it was and used an exponential model to estimate the growth rate. So, your first sentence should be revised (or deleted) with that in mind. Delete “during that time”. See comments below for L 313-315 for comments about the analysis and presentation of results.
L 302 – 305 – these results come out of the blue and I wonder what they mean for your analysis. Is this something that should be indicated in the methods and then you explain how you deal with it in regards to your analysis?
L 313-315 – I am having a hard time understanding your growth rate analysis. First, it would be more appropriate to present the estimate and 95% confidence intervals of the growth rate for each dose level rather than the estimate and SD current in Table 1. Given that the relationship is significant (F statistic on L 312-313) it should be obvious which dose level(s) do not overlap the estimate and SE of the control. Also, how does a “relatively low growth rate in the control group” mean for your ability to determine which exposure level significantly depresses growth rates? Second, I wonder if a generalized linear model might be more appropriate for this analysis using Dose as a continuous predictor variable. These same comments apply to comments made above in L 277-281.
L 317-322 – I recommend this paragraph be moved to the Methods section.
L 325 – write out “from leaves to larvae to adults”.
L 324 – 334 – I think the results on lines 327-334 are important to present so I recommend you move the first two sentences to the Methods section and then expand what these ratios indicate (it is too limited here to fully appreciate what these larval body burdens mean) and how you conduct this analysis – an important aspect to address in the Methods is how the response values (larva:leaf ratio) are not independent from the predictor values (leave values). In the results, please present the statistical results from the model presented in Figure 7.
L 343 – Experiment 1 also followed recommended dosages on the label so might be worth noting that recommended dosage of pesticides are unlikely to have an effect on growth, survival, mass, etc.
L 347 – suggest removing μg/kg so all units are consistent among these comparisons.
L 391 – the effect was not evident because you did not find a significant relationship. Please revise this sentence.
L 409 – please write out what is in the parentheses so that those without a background in pesticide applications (like me) can understand.
Figure 1 – please indicate what MDL stands for. Could probably exclude the 3 levels that had no data from this figure and indicate in the text that there was no detections from those doses. In the figure legend the sentence that presents the Chi-square, df and p-values is not clear and might be better if you state the results for leaves in one sentence and the results for larvae in another sentence – also present the results for leaves first since that is how the figure is laid out.
Figure 2 – I think if you only used data until day 9 (or 10) to estimate the growth rate (k) then you should exclude the lines beyond those days. Also, lines with different colours and a legend would help with interpretation here, or else use different shades of grey, or different line types (dashes, dots, etc.).
Figure 4 – please write out the x-axis what “ww” stands for.
Figure 6 – Similar comments as to presentation in Figure 2. Also, it would be good if the presentation/explanation of C and Ls were written in a consistent manner between Figure 2 and 6 to help readers interpret these data.
Figure 7 – Please add the 95% confidence interval around the line on this figure. Also, the caption is rather short. Since the points come from across a range of doses, it would be nice if the point color varied on a grey scale based on their dosage.
Table 1 – Please change “Average” to “Mean” throughout. In footnote 1 your left side of equation should At and “Exposure Study” should be changed to “Experiment” for consistence. In footnote 4 please revise the df which is indicated in two spots but without equal signs so I don’t know what it means.

---

## Round 0.2 · Major Revisions

Please revise the paper to focus more on the residue information. The toxicology data is weaker and perhaps can be treated as the reviewer suggests instead.

Reviewer 1 ·

Basic reporting

Basic reporting. Adequate. With some minor points:
1. Line 87: While systemic exposure of milkweed can result in internal neonic concentrations that become a dietary source of exposure, not aware label insecticide label claims that efficacy against insects is also made through contact exposure. A citation to support this route of exposure and/or a study showing systemic exposure results in external residues of neonic recommended.
2. In the results include the average milkweed ng/g with, or instead of, using Dose 1, Dose 2 etc. Easier for the reader to synthesize information by ng/g rather than Dose 1 etc. since the concentrations are very different across the 3 experiments.

Experimental design

Experimental design. The paper includes Experiments 1 and 2 that were described in the original submission with the addition of a third experiment that uses a higher exposure range. As noted by the authors, the first two experiments were confounded because of stress on the swamp milkweed, due at least in part to an aphid infestation. In the third experiment plants were maintained ‘aphid free.’
Line 118 – provide source of seeds
Line 138 – provide vendor name and location
Line 162 - how was the transfer accomplished? Same technique as described in experiment 2?

Line 171 – per plant or per replicate; i.e., after transferring larvae from the plant that had the attached cup with the eggs.
Line 223 - where were the larvae placed on the plants. assume larvae were placed on all 3 plants in a replicate. In Experiment 1 states btwn 1 and 8 larvae per plant. what was the range in larvae per plant for experiment 2?

Line 238 – clothianidin free, not pesticide free

Validity of the findings

Validity of Findings. The mortality and growth data is difficult to reconcile across the experiments. Experiment 2 is clearly difficult to interpret with the aphid infestation. Results from experiment 3 suggest the larvae were 4 -5 times more sensitive than larvae in Experiment 1 and Experiment 2, which is counter intuitive since plants in experiment 3 were, presumably, not stressed. In addition, mortality data from experiment 3 does not reflect a strong dose-response relationship – from 54 to 1007 ng/g, mortality was roughly 50% and was 100% at 1545 ng/g. Not surprisingly, the fit to the dose response curve must have been poor given the 95% CI interval ranged from about 10 to 450 ng/g. Perhaps an ANOVA analysis of mortality with post-hoc multiple comparisons would be more insightful? Are the mortality responses not significantly different across the 54 to 1007 ng/g groups but different than the response associated with the 1545 ng/g exposure? Unfortunately results from experiment 1, which might be able to inform mortality responses between control and 54 ng/g in experiment 3, do not seem to help in that in experiment 1 there was no larval mortality at a dose of 53.9 ng/g.

The residue data from soil, to plant tissue, to larvae to adults is the strength of the paper and provides some useful insights for researchers and risk assessors to consider when evaluating studies that use different exposure matrices.

Line 478 – The formulated product was selected to facilitate the means of treating the soil over a wide range of concentrations; i.e, the study was not undertaken to assess this specific formulation. Suggest this sentence be dropped. The text in line 485 is good.

Line 550 – This summary is misleading in that it implies a more precise summary of the toxicity studies than can be supported when looking at across and within experiment variability. It would be more appropriate to state that toxicity values were variable but generally well above high end clothianidin foliage concentrations of X to Y reported to date.

Some additional minor notes:

Table 3 – Define NA mean
Table 3, Experiment 2, leaf concentrations – include meaning of a, b, c footnotes
Table 3, Experiment 3. Could a post-hoc analysis; e.g., dunns-test, be used to determine if dose groups 4&5 different than 3, which may be different than 1&2?

Additional comments

As noted above the strength of the paper is the residue data and linking relationships of the residue data from soil, to leaves to larvae and adults. The paper’s toxicology data is very difficult to interpret and it is difficult to place high confidence in the results from a quantitative perspective. It would be reasonable use the toxicity data in a qualitative manner to conclude that it is unlikely currently reported clothianidin residue levels in plants sampled near agriculture fields are high enough to cause significant effects on survival or growth of monarch larvae and pupa.

---

## Round 0.3 · accepted · Accept

Thank you for addressing the reviewer's comments.